# Neuroimaging and Neurocognitive Outcomes in Older Patients with Multiple Myeloma Treated with Chemotherapy and Autologous Stem Cell Transplantation

**DOI:** 10.3390/cancers15184484

**Published:** 2023-09-08

**Authors:** Denise D. Correa, Behroze A. Vachha, Raymond E. Baser, Adrian Koch, Phillip Wong, Suril Gohel, Sergio Giralt, James C. Root

**Affiliations:** 1Department of Neurology, MSKCC—Memorial Sloan Kettering Cancer Center, New York, NY 10065, USA; 2Department of Neurology, Weill Cornell Medical College, New York, NY 10065, USA; 3Department of Radiology, UMass Chan Medical School, Worcester, MA 01665, USA; 4Department of Epidemiology & Biostatistics, Memorial Sloan Kettering Cancer Center, New York, NY 10065, USA; 5Department of Immune Monitoring Facility, Memorial Sloan Kettering Cancer Center, New York, NY 10065, USA; 6Department of Heath Informatics, Rutgers University School of Health Professions, Newark, NJ 08854, USA; 7Department of Medicine, Memorial Sloan Kettering Cancer Center, New York, NY 10065, USA; 8Department of Psychiatry & Behavioral Sciences, Memorial Sloan Kettering Cancer Center, New York, NY 10065, USA; 9Departments of Psychiatry, Weill Cornell Medical College, New York, NY 10065, USA

**Keywords:** multiple myeloma, neurocognitive, MRI, resting state functional connectivity, cytokines

## Abstract

**Simple Summary:**

We studied cognitive (thinking) abilities and brain structure and function in older adults with multiple myeloma—a cancer of plasma cells—treated with high-dose chemotherapy and a stem cell transplant. The initial results suggested that after the chemotherapy and transplant, functional connectivity was diminished in regions involving the frontal and parietal lobes of the brain, while brain structure and cognitive function remained relatively stable. We also found increases in markers of inflammation after the transplant. The findings provide supporting evidence for the vulnerability of frontal and parietal brain regions to the side effects of chemotherapy. These preliminary findings would support the design of large future studies with the goal of developing therapeutic interventions.

**Abstract:**

There is a paucity of research on treatment-related neurotoxicity in older adults with multiple myeloma (MM) treated with high-dose chemotherapy (HDC) and autologous SCT (HDC/ASCT), despite the increasing use of this regimen. We examined resting state functional connectivity (RSFC), gray matter (GM) volume, neurocognitive function (NF), and proinflammatory cytokines (PCy) in older patients with MM pre- and post-HDC/ASCT. Eighteen patients underwent MRI, NF tests, and serum PCy measurements prior to HDC/ASCT, and fifteen patients completed a follow up five-months post-HDC/ASCT. There were significant decreases in RSFC post-HDC/ASCT in (1) the central executive network (CEN) involving the *left* dorsolateral prefrontal cortex and *right* posterior parietal cortex (*p* = 0.022) and (2) the CEN involving the *right* posterior parietal cortex and the salience network involving the *right* dorsal anterior cingulate cortex (*p* = 0.029). There were no significant changes in GM or NF, except for improvements in attention (Digit Span Backward, *p* = 0.03). There were significant increases in several PCy post-HDC/ASCT (*p* ≤ 0.05). In conclusion, RSFC decreased in frontal, parietal, and cingulate cortices post-HDC/ASCT, NF was relatively stable, and several PCy increased. These findings are congruent with other studies in cancer patients and provide supporting evidence for the vulnerability of frontoparietal regions to chemotherapy’s adverse effects.

## 1. Introduction

There is compelling evidence that chemotherapy is associated with neurotoxicity [1], with suggested mechanisms including demyelination, microglia activation, immune dysregulation, and stimulation of proinflammatory cytokines (PCy) [2]. Reductions in prefrontal gray matter (GM) volume [3] and changes in resting state functional connectivity (RSFC) [4] have been reported post-chemotherapy in patients with breast cancer. We reported reductions in prefrontal GM volume and changes in white matter (WM) integrity in patients with hematological malignancies treated with high-dose chemotherapy (HDC) ± total body irradiation (TBI) and stem cell transplantation (SCT) [5,6]. Systematic reviews have indicated that neurocognitive dysfunction is prevalent both pre- and post-SCT [7].

Neurocognitive dysfunction has been documented in multiple myeloma (MM) patients after HDC and pre-autologous SCT (ASCT), with declines post-HDC/ASCT [8]; however, other studies have reported improvements in neurocognitive function (NF) months post-chemotherapy [9,10]. Pro-inflammatory cytokines (PCy) play a critical role in tumor growth and progression in MM, with high levels identified in many patients [11]. Aging can be associated with PCy elevations [12], suggesting that older MM patients may be even more susceptible to cytokine dysregulation related to disease and treatment, which may contribute to neuroinflammation.

There is a paucity of research investigating neurotoxicity in older MM patients undergoing HDC/ASCT, even though this intervention has been used more often in the elderly, and NF has been recognized as a critical dimension of survivorship in older cancer patients [13]. In this pilot study, we assessed GM volume, RSFC, NF, and PCy in older MM patients prior to HDC/ASCT and an average of five months post-HDC/ASCT.

## 2. Methods

### 2.1. Patients

MM patients scheduled for conditioning HDC/ASCT were recruited through the Adult Bone Marrow Transplant Service at Memorial Sloan Kettering Cancer Center (MSK). Eligibility criteria: (1) MM diagnosis, (2) complete, partial, or very good partial disease remission at enrollment, as per the standard International Myeloma Working Group Criteria, (3) aged 60–75 at enrollment, and (4) fluent in English. Exclusionary criteria: (1) disease progression during the study period, (2) CNS disease, or (3) history of neurological, psychiatric, or substance abuse disorders.

### 2.2. Measures

**Structural and Functional Imaging.** Patients were imaged in Tesla scanners (GE, Discovery 750 W, USA with a GEM HNU 24-channel head coil) at MSK; five patients were imaged in two different Tesla scanners at each timepoint using the same parameters. *Structural Imaging:* T1-weighted anatomical images with whole-brain coverage were obtained with spoiled gradient-recalled and high-resolution three-dimensional magnetization-prepared rapid acquisition with gradient-echo sequences. *Functional Imaging*: For rsfMRI, T2*-weighted images were acquired with a single-shot gradient echo-planar imaging (EPI) sequence (TR/TE = 2500 ms/30 ms, FA = 80°, slice thickness = 4 mm, matrix = 64 × 64). For the rsfMRI, patients were instructed to keep the eyes open and fixated on a crosshair.

Image Processing. For *structural image processing*, VBM analysis was performed using the longitudinal processing stream in the VBM8 toolbox (http://dbm.neuro.uni-jena.de/vbm/, accessed on 3 September 2023) under the SPM8 software package (Version 8, Wellcome Department of Imaging Neuroscience, London, UK) within MATLAB (Version 7, Mathworks, Inc., Natick, MA, USA). Following reconstruction, follow-up MPRAGE structural images were registered to baseline MPRAGE images for each subject, bias corrected, segmented into GM, WM, and cerebrospinal fluid compartments using the Montreal Neurologic Institute (MNI) T1-weighted template and tissue probability maps—linear and non-linear registered to MNI space—and the resulting GM tissue class smoothed using an isotropic Gaussian spatial filter (FWHM = 8 mm). *For rsfMRI pre-processing*, a data pre-processing scheme was implemented according to published methods [14]. Briefly, the first five timepoints of the fMRI data were removed to allow for T1 relaxation effects, followed by head-motion correction, co-registration, segmentation, normalization to MNI standard space, temporal regression of 24 head-motion parameters [15] and five principal components of WM and cerebral spinal fluid time series [16], temporal filtering between 0.01 to 0.1 Hz, and spatial-smoothing with a 6 mm full-width-at-half-maximum Gaussian filter. Using the head motion parameters, we calculated subject-specific measures of mean frame-wise displacement [17].

**Neurocognitive Tests and Self-Report Scales.** Patients completed standardized neurocognitive tests [18] in domains with documented sensitivity to cancer therapy adverse effects [19], including attention, executive function, and verbal learning and retrieval, as well as mood/fatigue self-report scales on the same day or within two weeks of the MRIs.

*Attention and Working Memory*: Longest Digit Span Forward-LDSF; Longest Digit Span Backward-LDSB; and Longest Number Sequencing-LSS (WAIS-IV) evaluate auditory attention and involve the repetition of a series of numbers forwards, backwards, and from lowest to highest; the Brief Test of Attention (BTA) assesses auditory selective attention; the Auditory Consonant Trigrams Test (ACT) assesses auditory attention and susceptibility to the effects of interference.

*Executive Functions*: Trail Making Test Parts A and B (TMTA and TMTB) assess timed visual scanning, graphomotor speed, and set-shifting; the Controlled Oral Word Association Test (COWA) assesses timed phonemic verbal fluency.

*Verbal Memory*: The Hopkins Verbal Learning Test—Revised: Total, Delayed Recall, Discrimination Index (HVLT-R-T; HVLT-R-D; HVLT-R-DI) is a test of verbal memory. It requires the learning and recall of a word list over three trials and after a delay, and recognizing the words in a forced-choice format.

*Self-Report Scales*: The Center for Epidemiological Study-Depression (CES-D) [20] test involves the rating of perceived symptoms of depression; the Functional Assessment of Chronic Illness Therapy-Fatigue Subscale, Version 4 (FACIT-FS V-4) [21] involves the rating of perceived symptoms of fatigue.

**Multiplex Cytokine Panel.** Blood samples were collected pre- and post-HDC/ASCT on the same day as the neurocognitive assessment and delivered to the Immune Monitoring Core Facility at MSK for plasma isolation and frozen storage until ready for batch analysis. PCy were quantitated from thawed plasma samples following the manufacturer’s instructions for the V-PLEX Human Proinflammatory Panel 10-plex kit (Meso Scale Diagnostics-MSD, Cat #K15049D-1), which included interleukin 1beta (IL-1β), interleukin 2 (IL-2), interleukin 4 (IL-4), interleukin 6 (IL-6), interleukin 8 (IL-8), interleukin 10 (IL-10), interleukin 12 (IL-12), interleukin 13 (IL-13), interferon gamma (IFNγ), and tumor necrosis factor alpha (TNF-α).

## 3. Statistical, Imaging, and Cytokine Analyses

*Voxel-Based Morphometry (VBM).* Following omnibus testing, pairwise *t*-tests were performed at the group level to analyze within-group changes from pre- to post-HDC/ASCT. For the structural contrast, the initial uncorrected voxel-wise threshold was *p* ≤ 0.001, with resulting maps family-wise errors corrected over the whole brain at *p* ≤ 0.05.

*Resting State Functional Connectivity Analysis (RSFC)* was performed using region-of-interest-based correlations, as described previously [22]. Three resting state networks (RSNs) were extracted for prioritized analyses: the central executive network (CEN), the salience network (SN), and the default mode network (DMN). Spherical regions of interest (ROIs) were created surrounding the ROI coordinates appropriate to the RSNs of interest. The CEN and SN ROIs were created using the coordinates defined by Uddin et al. [23] and Yang et al. [24], respectively. The DMN ROIs were created using NeuroSynth [25] with the functional connectivity and co-activation map derived in 02/2022 using the term “default mode”. For each of the coordinates, a 6 mm spherical ROI was created. Table 1 lists the MNI coordinates for each ROI.

Correlation matrices were produced by extracting the time course from each of the ROIs and computing the Pearson correlation coefficient (*r*) between each ROI pair in the CEN, SN, and DMN. Each of the pair-wise ROI correlations were Fisher z-transformed for further statistical analysis. Changes in RSFC z-scores from pre- to post-HDC/ASCT were assessed using linear mixed models, adjusting for the scanner (1, 2).

*Neurocognitive Analysis*: Raw neurocognitive test scores were transformed into z-scores based on age-corrected published normative values. Neurocognitive test z-scores and self-report scale scores were summarized at each timepoint using descriptive statistics, and differences in scores from pre- to post-HDC/ASCT were compared using Wilcoxon signed rank tests. Standardized effect sizes (i.e., Cohen’s *d*) were calculated to quantify the magnitude of score changes over time. The false discovery rate (FDR) was used to adjust *p*-values for multiple comparisons. The Reliable Change Index (RCI), which represents the change in scores divided by the standard error of measurement, was used to identify patients whose raw scores improved or declined beyond expected levels due to practice effects and measurement errors. For each test score, we calculated the proportion of patients with RCI-indicated reliable decline.

### Cytokine Panel Analysis

Cytokine data was analyzed using the MSD Discovery Workbench^®^ software to measure the levels of a ten-PCy panel at each timepoint. A four-parameter logistic (4PL) fit calibration curve was generated for each analyte using the standards to calculate the concentration of each analyte. The upper and lower limits of quantitation for each PCy were established as the highest and lowest points of the standard curve on each plate whose back calculated values were within 80–120% of the expected values of the 4PL regression fit and exhibited less than 20% CV across the duplicate standard wells.

**Correlations.** Spearman correlations were calculated to assess the association of RSN z-scores, neurocognitive test z-scores, self-report scale scores, and PCy levels separately for each timepoint.

## 4. Results

Eighteen MM patients completed a neurocognitive assessment and a brain MRI pre-HDC/ASCT, and fifteen patients were available for follow up for an average of five months (median = 5.82, range = 3.50–7.00) post-HDC/ASCT. One patient was excluded from the imaging analysis due to scan misregistration at follow up. Thirteen patients provided blood samples for PCy analysis at each timepoint. Descriptive statistics for demographic and disease variables are presented in Table 2. All patients received conditioning HDC the day before the ASCT. Five patients were treated with Siltuximab, an IL-6 blocker (median half-life elimination: ~21 days, range:14 to 30 days) [26,27], infused seven days pre-HDC/ASCT and twenty-one days post-HDC/ASCT, as part of a separate MSK protocol investigating cytokine response during the acute phase of ASCT (IRB #17-365), which did not require high IL-6 levels for enrollment. In this study, data collection occurred prior to the first Siltuximab infusion and at least 3–4 months after the second infusion for all patients. Among the three patients who did not return for follow up, one had disease progression, one was deceased, and one relocated.

### 4.1. Structural and Functional Imaging

The results showed significant decreases in RSFC in (1) CEN ROIs involving the *left* dorsolateral prefrontal cortex (L-DLPFC) and *right* posterior parietal cortex (R-PPC; *p* = 0.022), and (2) the CEN ROI involving the R-PPC and the SN ROI involving the *right* dorsal anterior cingulate cortex (R-dACC; *p* = 0.029) from pre- to post-HDC/ASCT (Figure 1); these comparisons were no longer significant after correction for multiple comparisons. There were no significant changes in RSFC in the DMN. The VBM analysis results showed no significant changes in regional GM volumes (FWE corrected, *p* > 0.05).

### 4.2. Neurocognitive Function and Self-Report Scales

The neurocognitive test z-scores and self-report scales scores are presented in Table 3. Mean z-scores were within the average range for all tests, except for ACT-perseverations. There were no significant changes in test scores from pre- to post-HDC/ASCT, except for a significant improvement in attention (Longest Digit Span Backward, *p* = 0.03); this comparison was no longer significant after multiple comparison correction. There were no significant changes in the CES-D and FACIT-FS scores, and scores were within normal limits at each timepoint [20,28]. There were no significant differences in the neurocognitive tests between the fifteen patients who completed both timepoints and the three patients who performed the pre-HDC/ASCT assessment only.

The RCI results showed that most patients (>67%) had no reliable change in neurocognitive tests from pre- to post-HDC/ASCT. Patients showed reliable declines on the HVLT-R-D (7%), TMT A (34%), TMT B (13%), and BTA (7%). Reliable improvements were seen with the HVLT-R-T (13%), HVLT-R-D (7%), TMT A (34%), TMT B (20%), and BTA (27%).

### 4.3. Multiplex Cytokine Panel

PCy levels were generally low in most patients at both timepoints, and pre-HDC/ASCT values were below the limits of quantitation for IL-1b, IL-2, IL-4, IL-10, IL-12, and IL-13. There were significant increases in PCy post-HDC/ASCT, including IL-1b, IL-2, IL-4, IL-8, IL-12, IL-13, and TNFα (*p* ≤ 0.05, FDR-corrected). Table 4 includes PCy medians and ranges. The five patients treated with Siltuximab exhibited high IL-6 levels post-HDC/ASCT and showed increases in IL-1b, IL-2, IL-4, IL-12, and IL-13 levels, but the comparisons were not statistically significant. There were no significant differences in the neurocognitive tests or self-report scales between patients with high versus normal IL-6 levels post-HDC/ASCT.

**Correlations**. There were no significant correlations among RSN z-scores, neurocognitive test z-scores, self-report scale raw scores, and PCy levels either pre- or post-HDC/ASCT.

## 5. Discussion

This is the first pilot study describing alterations in RSFC in older MM patients treated with HDC/ASCT. The preliminary results showed decreased connectivity in the CEN involving the L-DLPF and R-PPC and in the CEN R-PPC and the SN R-dACC from pre- to an average of five months post-HDC/ASCT, suggesting that the adverse effects of HDC may be of concern in this population. The CEN has been described as a frontoparietal network involved in attention control, working memory, and processing speed [29,30]. The SN includes the dACC and anterior insula (AI) cortex and is involved in the monitoring and processing of errors and conflict [31], as well as attention control in the presence of distractions [32]. Interestingly, these results are in overall agreement with evidence of impaired performance on a neurocognitive test assessing susceptibility to interference. Although we found no significant changes in the DMN, the CEN and SE results were consistent overall with breast and ovarian cancer studies, suggesting that frontoparietal regions are susceptible to chemotherapy adverse effects [33,34].

A structural neuroimaging literature review on non-CNS cancer patients [35] described consistent findings of reduced GM volume mostly in frontal, temporal, and parietal regions, as well as diffuse alterations in WM integrity—mostly in patients treated with chemotherapy. Our study in patients with hematologic malignancies treated with HDC ± TBI and SCT [5] showed GM volume reductions in the bilateral middle frontal gyrus from pre- to one-year post-SCT. However, the current results showed no significant changes in regional GM volume post-HDC/ASCT, suggesting no significant adverse effects on brain structure. However, longer follow up periods may be required to detect treatment-related delayed adverse effects on brain structure.

It is estimated that at least 50% of patients with hematological malignancies experience neurocognitive dysfunction prior to SCT, with either stable performance or declines in the months to years post-SCT [36,37]. In this study, NF was within the average range—although below expected levels considering the mean education of the cohort, with impairment on a test of susceptibility to interference/working memory. This could be in part related to the disease and the residual adverse effects of proteasome inhibitors, chemotherapy, and immunotherapy, [38] and in some patients, to the side effects of dexamethasone [39] and lenalidomide [40]. NF remained relatively stable, with improvements in attention, suggesting no significant adverse effects from HDC/ASCT; however, it has been suggested that stable NF post-SCT may represent a lack of improvement, expected due to practice effects [41]. Self-report scales indicated no depression or fatigue, or changes from pre- to post-HDC/ASCT. There is a limited number of neurocognitive studies focusing on MM patients. Some studies have reported improvement in NF following chemotherapy in older MM patients (Bury-Kaminska, 2021) [9] and one-year post-SCT in a sample of mostly MM patients (Jacobs et al., 2007) [10]. However, neurocognitive dysfunction has been reported in MM patients (mean age = 58 years, SD = 8.2) after HDC and pre-ASCT, with declines one and three months post-HDC/ASCT [8].

Cytokine dysregulation is common in MM and may influence the development of adverse effects [11]. High PCy levels may interfere with the blood–brain barrier [42] and may induce the proliferation of microglia and CNS inflammation [43]. Increased PCy levels have been associated with neurocognitive dysfunction [44] and with changes in brain structure [45] in breast cancer patients. Neurocognitive dysfunction is associated with high IL-6 levels in patients with leukemia and myelodysplastic syndrome [46]. However, the mechanisms of chemotherapy-related PCy dysregulation and NF are not well understood, with conflicting results on the strength and direction of these associations [36,47]. We observed low PCy levels at both timepoints, with significant increases post-HDC/ASCT—possibly influenced by HDC adverse effects [48]. In some patients, the effects of dexamethasone and lenalidomide may have also influenced PCy levels. In the patients treated with Siltuximab, IL-6 and other PCy levels increased post-HDC/ASCT. There is evidence that an IL-6 blockade may result in a paradoxical increase in IL-6 and systemic inflammation [49,50]; however, there were no significant associations with the neurocognitive and neuroimaging outcomes. Additional research would be required to reconcile our preliminary findings with the role of MM and its treatment on PCy dysregulation and treatment-related neurotoxicity.

In this pilot study, changes in RSFC were more pronounced than in NF—possibly reflecting the use of compensatory mechanisms to maintain cognitive performance in the context of decreased CEN and SE connectivity [51], as well as the greater sensitivity of advanced neuroimaging tools in detecting subtle alterations in functional connectivity in vulnerable regions. However, the RSFC results were no longer significant after multiple comparison correction, and there were no significant associations between NF, RSFC, and PCy levels. These findings may be in part related to the small sample size and reduced statistical power for detecting small changes in RSFC and associations among variables beyond what was reported. In addition, MM patients receive fewer lines of prior chemotherapy and less neurotoxic agents and conditioning regimens, which may be associated with less-pronounced adverse effects compared to other hematologic malignancies [41,52]. The small sample size may impact the generalizability of these findings to other MM patients and populations.

## 6. Conclusions

Decreased RSFC in older MM patients following HDC/ASCT provides further evidence for the prevailing notion that frontal-parietal regions may be vulnerable to chemotherapy adverse effects. Longitudinal studies with larger sample sizes and longer follow-up periods are needed to further investigate the neural correlates and trajectory of chemotherapy-related neurotoxicity, as well as the role of PCy in older MM patients. The increasing number of older patients treated with stem cell transplant underscores the importance of investigating treatment adverse effects and identifying patients at increased risk. These studies will guide the development and implementation of targeted interventions to prevent or minimize neurocognitive dysfunction and its impact on quality of life.

## Figures and Tables

**Figure 1 cancers-15-04484-f001:**
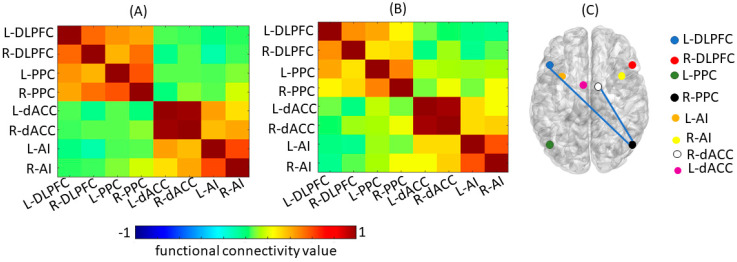
(**A**) Mean Functional Connectivity for Central Executive Network and Salience Network pre-ASCT. (**B**) Mean Functional Connectivity for Central Executive Network and Salience Network post-ASCT. (**C**) Group level difference in Functional Connectivity between pre- and post-ASCT. Blue lines represent significantly decreased connectivity (*p* < 0.05) from pre- to post-ASCT. DLPFC = Dorsolateral Prefrontal Cortex, PPC = Posterior Parietal Cortex, dACC = Dorsal Anterior Cingulate Cortex, AI = Anterior Insula; L = Left, R = Right.

**Table 1 cancers-15-04484-t001:** MNI Coordinates.

Network Name	Region Name	X	Y	Z
Default Mode Network				
	Posterior Cingulate Cortex	−2	−54	26
	Medial Prefrontal Cortex	2	50	−6
	L-Angular Gyrus	−50	−62	32
	R-Angular Gyrus	46	−70	32
Salience Network				
	L-Dorsal Anterior Cingulate Cortex	−5	26	31
	R-Dorsal Anterior Cingulate Cortex	5	26	31
	L-Anterior Insula	−34	15	−4
	R-Anterior Insula	37	20	−6
Central Executive Network				
	L-Dorsolateral Prefrontal cortex	−46	20	44
	R-Dorsolateral Prefrontal cortex	46	20	44
	L-Posterior Parietal Cortex	−40	−56	44
	R-Posterior Parietal Cortex	52	−52	50

L = Left; R = Right.

**Table 2 cancers-15-04484-t002:** Demographic Characteristics and Treatment History (*n* = 18).

Demographics	
Sex (M/F)	10/8
Handedness (R/L)	17/1
Education (years)	
Mean (SD)	14.33 (3.58)
Median (Range)	14.5 (13–18)
Age at study entry (Years)	
Mean (SD)	66.11 (3.60)
Treatment regimen pre-ASCT	
RVd	5 (28%)
KRd	5 (28%)
Lenalidomide (alone)	4 (22%)
KRd + RVd	1 (5.5%)
CyBorD ± KRd	2 (11%)
CyBordD + KRd + RVd	1 (5.5%)
Response to pre-ASCT treatment	
CR	7 (39%)
VGPR	9 (50%)
PR	2 (11%)
Time since pre-ASCT treatment	
0–1 months	11 (61%)
2 months	5 (28%)
>2 months	2 (11%)
ASCT Conditioning Regimen	
Melphalan—Single Dose	17 (95%)
Melphalan—Two-Dose	1 (5%)
Time from Baseline * to ASCT	
≤1 month	15 (83%)
1.1–4 months	3 (17%)
Relevant Medications	
Pre-ASCT	
Dexamethasone	6 (33%)
Lenalidomide	4 (22%)
Post-ASCT	
Dexamethasone	0 (0%)
Lenalidomide (maintenance)	3 (17%)

SD = standard deviation, CR = complete response, VGPR = very good partial response. PR = partial response; KRd = carfilzomib/lenalidomide/dexamethasone, RVd = lenalidomide/bortezomib/dexamethasone, CyBorD = cyclophosphamide/bortezomib/dexamethasone. Baseline * = pre-ASCT neurocognitive, MRI, blood sample collection, relative to transplant date.

**Table 3 cancers-15-04484-t003:** Neurocognitive Test Z-Scores and Self-Report Scale Raw Scores.

Measures	Pre-ASCT *n* = 15 Mean (SD)	Post-ASCT *n* = 15 Mean (SD)
Attention/Working Memory		
LDSF	0.16 (1.18)	0.20 (1.21)
LDSB	−0.02 (0.96)	0.50 (1.09) *
LLSS	−0.14 (1.05)	0.17 (1.14)
BTA	−0.07 (1.07)	0.33 (0.97)
ACT-T	−0.24 (1.02)	−0.46 (1.10)
ACT-P	−1.60 (1.41)	−1.64 (1.55)
Executive Functions		
TMT A	−0.35 (0.80)	−0.05 (1.01)
TMT B	−0.71 (1.03)	−0.16 (1.13)
COWA	−0.94 (1.17)	−0.51 (1.08)
Verbal Memory		
HVLT-R-T	−0.63 (1.09)	−0.49 (0.93)
HVLT-R-D	−0.58 (0.98)	−0.55 (1.42)
HVLT-R-DI	−0.02 (0.86)	0.34 (0.69)
Self-Report Scales		
CES-D	9.67 (5.49)	9.73 (6.37)
FACIT-FS	38.89 (7.55)	39.40 (6.72)

LDSF = Longest Digit Span Forward; LDSB = Longest Digit Span Backward; LNS = Longest Number Sequencing; BTA = Brief Test of Attention; ACT-T = Auditory Consonant Trigrams—Total Score; ACT-P = Auditory Consonant Trigrams—Perseverations; TMTA = Trail Making Test A; TMTB = Trail Making Test B; COWA = Controlled Oral Word Association Test; HVLT-R-T = Hopkins Verbal Learning Test-Revised-Total Learning; HVLT-R-D = Hopkins Verbal Learning Test-Revised-Delay; HVLT-R-DI = Hopkins Verbal Learning Test-Revised- Discrimination Index; CES-D = Center for Epidemiological Study-Depression; FACIT-FS = Functional Assessment of Chronic Illness Therapy Version IV-Fatigue Subscale. * *p* = 0.03.

**Table 4 cancers-15-04484-t004:** Cytokine Levels (pg/mL).

Cytokines	Pre-ASCT *n* = 13 Median (Range)	Post-ASCT *n* = 13 Median (Range)
IL-1β	0.00 (0.00–0.00)	0.00 (0.00–0.18) *
IL-2	0.08 (0.01–0.28)	0.35 (0.19–1.05) *
IL-4	0.01 (0.00–0.02)	0.01 (0.00–2.53) *
IL-6	0.78 (0.65–0.86)	2.30 (0.58–1572)
IL-8	4.53 (3.41–6.62)	7.70 (6.75–10.96) *
IL-10	0.19 (0.12–0.42)	0.28 (0.15–0.48)
IL-12	0.05 (0.01–0.14)	0.10 (0.08–3.03) *
IL-13	0.00 (0.00–0.13)	0.25 (0.00–0.88) *
IFNγ	2.39 (1.82–6.91)	4.15 (2.43–7.38)
TNFα	0.56 (0.44–0.75)	0.94 (0.60–1.04) *

Pg/mL = picograms per milliliter; IL-1 β = interleukin 1 beta, IL-2 = interleukin 2, IL-4 = interleukin 4, IL-6 = interleukin 6, IL-8 = interleukin 8, IL-10 = interleukin 10, IL-12 = interleukin 12, IL-13 = interleukin 13, IFNγ = interferon gamma, TNF-α = tumor necrosis factor alpha. * *p* ≤ 0.05 (FDR-corrected).

## Data Availability

Data will be available upon request to the corresponding author.

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
