# Peer review of "Neuroimaging and Neurocognitive Outcomes in Older Patients with Multiple Myeloma Treated with Chemotherapy and Autologous Stem Cell Transplantation"

_cancers, 2023, doi:10.3390/cancers15184484_

Round 1

Reviewer 1 Report

The manuscript by Correa et al is well-prepared and presents an interesting and valuable contribution to the understanding of neurocognitive changes in older adults with multiple myeloma following high-dose chemotherapy and autologous stem cell transplant. The authors have presented a thorough study with careful consideration of potential limitations. The reviewer has provided some minor comments below, to enhance clarity and ensure the accurate representation of all information.

 1. The study population's age range is between 60 and 75, making it important to address the potential confounding effects of age-related cognitive decline. Since neurocognitive changes can occur due to aging alone, it is necessary to discuss how the study controlled for these effects, especially in an older population.

2. The description of the neurocognitive tests administered to participants is quite concise. It would be helpful to provide more details about the rationale for selecting these specific tests and how they relate to the cognitive functions affected by chemotherapy, especially in the context of multiple myeloma.

3. The study involves a relatively small sample size (18 patients pre-ASCT and 15 post-ASCT). Given the complex nature of the analysis involving multiple variables, the statistical power should be explicitly addressed. Discuss the limitations of the study due to sample size and how these limitations might impact the generalizability of the findings.

4. The study did not find significant changes in regional gray matter volume. However, this result should be interpreted cautiously since structural changes might require longer follow-up periods to become evident. Additionally, the findings related to resting-state functional connectivity (RSFC) changes are significant but did not survive correction for multiple comparisons. It's important to discuss the implications of these findings, even though they were not fully statistically significant.

5. The finding of relatively stable neurocognitive function post-HDC/ASCT is interesting, especially considering the expected cognitive decline following chemotherapy in cancer patients. Discuss potential explanations for this stability, including the role of the specific treatment regimens used in multiple myeloma.

6. The significant increases in proinflammatory cytokines (PCy) post-HDC/ASCT are noteworthy. However, further discussion is needed to explain how these cytokine changes might contribute to the observed neurocognitive and functional connectivity alterations. It would also be beneficial to acknowledge the potential influence of medications (e.g., Siltuximab) on cytokine levels.

7. While the study's limitations are acknowledged in the conclusion, there could be a more comprehensive discussion about the clinical implications of the findings. How might these findings guide future therapeutic interventions? What are the potential implications for clinical practice and patient care?

Author Response

Please find attached a single Word document with the responses to all three reviewers.

Reviewer 2 Report

The authors studied the neuroimaging and neurocognitive changes in old myeloma patient underwent auto-SCT. The study has novelty and is interesting.

1. How about these neuroimaging and congnitive changes compared to young myeloma patient underwent auto-SCT? And also compared to other malignacies with SCT?

2. Does the changes comes from chemotherapy agents or others? How about the chemotherapy agents during SCT?

3. Does previous treatment agents affects these changes? Some anti-myeloma agents have neuro-toxicity, does it impact on these changes?

4. Does the changes persist or can be recovered?

Author Response

Please find attached a revised manuscript with tracked changes, in response to the reviewers comments.

Responses the reviewers were sent as a single Word document.

Reviewer 3 Report

This is an interesting paper evaluating the use of neuroimaging and pro-inflammatory cytokines in patients undergoing high dose chemo and stem cell transplant in myeloma. However, interpretation of the findings is difficult due to limitations by the small sample size and lack of significant changes in the structural and functional imaging results after correction for multiple comparisons, if the reader is following the authors' comment correctly, and there were no significant correlations among RSN z-scores etc pre and post HDC/ASCT?

I found the paper interesting but somewhat difficult to follow with the presentation of methods and results, in particular was unclear of the reference to time in table 2- time from baseline to ASCT- is time referred to here the time of the imaging? If so, there is considerable discrepancy with 17% of patients between imaged up to 4 months from baseline to ASCT, which would likely influence the results of PCy and imaging?

 I am also confused on the details provided in the treatment history as we are informed that patients are treated with various regimens ranging from RVd, KRd, lenalidomide, KRd, RVd, all regimens which contained lenalidomide (R) - so why is it only 4 patients (22%) considered to have received lenalidomide as relevant medications (Further down Table 2 where this is mentioned). Clarification of the use of lenalidomide post ASCT would also be prudent - is this used as maintenance/ consolidation which may also impact on results?

Details of the conditioning regimen can be made clearer- Melphalan single dose vs multiple dose- is the referring to the total dose been the same but split over 2 days, if so - why is this undertaken (?renal impairment or other factors that may potentially influence the levels of Pro-inflammatory cytokines).

THe use of Siltuximab as per another MSK protocol- is confounding, as siltuximab is not generally used in stem cell transplant for myeloma, and similarly, it would also be unusual for HDC to be given on the same day as stem cells return- so perhaps this point can be clarified.

Patients with myeloma are usually diagnosed at a median age of 60 so this study does capture the typical patient rather than "older patients"- the concept of cognition been impaired by chemo is certainly recognized in significant number of patients- which occasionally persist, but there are likely other factors at play- perhaps the authors may discuss this as this would make the paper more useful and relevant to the readership.

Overall this paper has merit but improvements should be made.

Author Response

Responses to reviewers were sent a single Word document.

Round 2

Reviewer 2 Report

The authors responded and modified the manuscript as reviewers' suggestion. I think it is suitable to be published in the journal.

Reviewer 3 Report

Improved paper